# Daidzein Intake Is Associated with Equol Producing Status through an Increase in the Intestinal Bacteria Responsible for Equol Production

**DOI:** 10.3390/nu11020433

**Published:** 2019-02-19

**Authors:** Chikara Iino, Tadashi Shimoyama, Kaori Iino, Yoshihito Yokoyama, Daisuke Chinda, Hirotake Sakuraba, Shinsaku Fukuda, Shigeyuki Nakaji

**Affiliations:** 1Department of Gastroenterology, Hirosaki University Graduate School of Medicine, Hirosaki 036-8562, Japan; chikaran0601@yahoo.co.jp (C.I.); donkeyra3@yahoo.co.jp (D.C.); hirotake@hirosaki-u.ac.jp (H.S.); sfukuda@hirosaki-u.ac.jp (S.F.); 2Aomori General Health Examination Center, Aomori 030-0962, Japan; 3Department of Obstetrics and Gynecology, Hirosaki University Graduate School of Medicine, Hirosaki 036-8562, Japan; iino-ka@hirosaki-u.ac.jp (K.I.); yokoyama@hirosaki-u.ac.jp (Y.Y.); 4Department of Social Medicine, Hirosaki University Graduate School of Medicine, Hirosaki 036-8562, Japan; nakaji@hirosaki-u.ac.jp

**Keywords:** equol, daidzein, gut microbiota, *Asaccharobacter celatus*, *Slackia isoflavoniconvertens*

## Abstract

Equol is a metabolite of isoflavone daidzein and has an affinity to estrogen receptors. Although equol is produced by intestinal bacteria, the association between the status of equol production and the gut microbiota has not been fully investigated. The aim of this study was to compare the intestinal bacteria responsible for equol production in gut microbiota between equol producer and non-producer subjects regarding the intake of daidzein. A total of 1044 adult subjects who participated in a health survey in Hirosaki city were examined. The concentration of equol in urine was measured by high-performance liquid chromatography. The relative abundances of 8 bacterial species responsible for equol production in the gut microbiota was assessed using 16S rRNA amplification. There were 458 subjects identified as equol producers. The proportion of equol production status and the intake of daidzein increased with age. Daily intake of daidzein was larger in equol-producer. The intestinal bacteria, which convert daidzein to equol were present in both equol producers and non-producers. However, the relative abundance and the prevalence of *Asaccharobacter celatus* and *Slackia isoflavoniconvertens* were significantly higher in equol producers than those in equol non-producers. The intestinal bacteria that convert daidzein to equol are present in not only the equol producers but also in the non-producers. The daidzein intake is associated with the equol production status through an increase of *A. celatus* and *S. isoflavoniconvertens* in the gut microbiota.

## 1. Introduction

Equol, a metabolite of the soy isoflavone daidzein produced by the gut bacteria, has been reported to have a great affinity to estrogen receptors [1]. Equol has been associated with decreased risks of breast, prostate, and colon cancers; cardiovascular diseases; osteoporosis; and hormone-dependent diseases [2,3,4,5,6]. However, an ability to produce equol varies among the individuals, because only people who possess the intestinal bacteria capable of producing equol are regarded as equol producers. The equol production status has been associated with the consumption of isoflavone daidzein. Previous studies have been demonstrated that only 20–30% of the people in Western countries are equol producers [7,8,9], while the prevalence of equol producers is 40–60% among people in Asian countries where the isoflavone daidzein is regularly consumed [3,10,11].

To date, more than 10 species of gut microbes have been recognized to be responsible for equol production. Several studies have investigated the relationship between the intake of isoflavone daidzein and the equol-production status [12,13,14,15]. However, these studies were regardless of the gut microbiota, which plays a significant role in the equol production. Furthermore, few studies have been conducted to examine the influence of daily intake of daidzein on the equol-production status considering the gut microbiota.

We compared the prevalence of 8 bacterial species in the gut microbiota that convert daidzein to equol between the equol producers and non-producers to investigate the association between the equol-production status and the equol-producing microbiota. We also assessed the daily intake of daidzein to evaluate whether the daily intake of daidzein is different between the equol-producers and non-producers.

## 2. Material and Methods

### 2.1. Study Subjects

There were 1118 adult participants in the Iwaki Health Promotion Projects held in June 2015, in Iwaki District of Hirosaki City located in north Japan (Figure 1). Of these, we excluded 11 subjects whose urine sample were not collected; 11 subjects whose urine concentrations of daidzein were under the detectable level; and 52 subjects whose stool samples were not collected. Finally, 1044 subjects (411 men and 633 women) were included.

### 2.2. Urine Concentration of Equol and Daidzein

The concentration of equol and daidzein in urine were measured by using a modified high-performance liquid chromatography method (HPLC) [16]. The equol production status was defined by a urinary log 10 - transformed equol/daidzein ratio of −1.75 or more as described previously [17].

### 2.3. Intake of Daidzein

The intake of daidzein was calculated based on a brief self-administered diet history questionnaire (BDHQ), a convenient diet assessment questionnaire developed in Japan [18,19,20]. The BDHQ included questions concerning the intake frequency of 58 food and beverage items commonly consumed in Japan. Those foods included three traditional Japanese soybean products: natto, tofu, and fried tofu.

### 2.4. Next Generation Sequence Analysis of Gut Microbiota

Fecal samples were collected in commercial containers (TechnoSuruga Laboratory Co., Ltd., Shizuoka, Japan) and were suspended in a guanidine thiocyanate solution (100 mM Tris-HCl (pH 9.0), 40 mM Tris-EDTA (pH 8.0), 4M Guanidine Thiocyanate). These samples were kept at −80 °C until DNA extraction. According to previous studies, a series of representative bacterial species in the human gut microbiota were analyzed using the primers for the V3 - V4 region of 16S rDNA of the prokaryotes [21,22]. The sequencing was conducted using an Illumina MiSeq system (Illumina, San Diego, CA, USA). The methods for quality filtering of the sequence was as follows: the only reads that had quality value scores of scores of ≥20 for more than 99% of the sequence were extracted for the analysis. Detection and identification of the bacteria from the sequences were performed using Metagenome@KIN software (World Fusion Co., Tokyo, Japan) and the TechnoSuruga Lab Microbial Identification database DB-BA 10.0 (TechnoSuruga Laboratory) at 97% sequence similarity.

We compared the relative abundance of the bacterial species in the gut microbiota between the equol producers and non-producers. The relative abundance is the percentage composition of the reads of a bacterium to the total reads of the gut microbiota. To estimate the contribution of each bacterium, the weighted average difference rank was calculated. The following 8 bacterial species that convert daidzein to equol were surveyed; *Adlercreutzia equolifaciens* [23], *Asaccharobacter celatus* [24], *Bacteroides ovatus* [25], *Finegoldia magna* [26], *Lactobacillus mucosae* [27], *Slackia equolifaciens* [28], *Slackia isoflavoniconvertens* [28], and *Streptococcus intermedius* [25].

### 2.5. Statistical Analysis

Statistical analyses of the clinical data were performed using the Statistical Package for the Social Science (SPSS) version 20.0 (SPSS Inc., Chicago, IL, USA) and the R software (R Foundation for Statistical Computing, version R-3.4.3). Categorical variables are shown as frequencies and percentages, and, continuous variables are shown as the mean with the standard deviation or the median with the interquartile range. The categorical variables were compared using the Chi-square test and the continuous variables were compared using the Student’s *t*-test or Mann-Whitney U-test. A *p*-value of less than 0.05 was considered significant for all the tests. Alpha-diversity was evaluated using Shannon index and chao1 while beta-diversity was evaluated by principal components analysis and statistically analyzed using permutation multivariate analysis of variance.

### 2.6. Ethics Statement

This study was performed in accordance with the ethical standards of the Declaration of Helsinki, and approved by the ethics committee at Hirosaki University Medical Ethics Committee (2014-377). All patients provided written informed consent for this study.

## 3. Results

Among the 1044 subjects, 458 (44%) were identified as the equol producers (Table 1). Although gender was not different between the equol producers and non-producers, there was a significant difference between the ages. Alpha-diversity analysis showed a higher gut microbiota complexity in the equol producers compared to those in non-producers (Figure 2). The gut microbiome was also significantly different between of equol the equol producers and non-producers by principal component analysis (*p* < 0.001) (Figure 3). Figure 4 shows the difference of the relative abundances of gut microbiota between the equol producers and non-producers at the phyla and genera level.

The proportion of the equol producers was increased with the age and more than half of the subjects in older than 60 years were the equol producers (Figure 5). Furthermore, the daily intake of daidzein was also higher in the equol-producers. The median intake of daidzein based on the BDHQ was 15.0 mg/day in the equol producers and was 12.8 mg/day in the non-producers (*p* = 0.004) (Table 1). The daily intake of daidzein was also increased with the age (Figure 6).

The relative abundance of *A. celatus* and *S. isoflavoniconvertens* was significantly higher in the equol producers than those in the equol non-producers (Table 2). The prevalence of *A. equolifaciens*, *A. celatus*, *B. ovatus*, *S. equolifaciens* and *S. isoflavoniconvertens* in equol producer was significantly higher in the equol producers than in the non-producers (Table 3). In the equol producers, the prevalence of *A. celatus* and *S. isoflavoniconvertens* was 50.2% and 38.9%, respectively, while these values were only 11.8% and 4.3% in non-producers, respectively in the non-produces. The analyses among different age-groups by decades revealed that the prevalence of *A. celatus* and *S. isoflavoniconvertens* in the equol producers were significantly higher than that in the non-producers in all the decade-groups (Figure 4). Furthermore, in the equol producers only, the relative abundance of *S. isoflavoniconvertens* tended to increase with the age.

## 4. Discussion

The current study demonstrated that not only the equol producers, but also the non-producers possess the bacteria that convert daidzein to equol. However, there were some differences in the structure of gut microbiota and the prevalence of bacteria that convert daidzein to equol. Particularly, the abundance of *A. celatus* and *S. isoflavoniconvertens* were significantly different between the equol producers and non-producers. Moreover, the daily daidzein intake was significantly higher in the equol producers as compared the non-producers.

This is the first study to investigate the differences in the intestinal bacteria, which produce equol from daidzein, comparing between the equol producers and the non-producers in a mass-survey including more than a thousand subjects. In this study, the prevalence of the equol producers was more than 40%, consistent with a previous Japanese study [3]. Moreover, the equol producers were significantly older than the non-producers. Similar to our study, some previous studies have also reported that the prevalence of the equol producers among the younger male subjects to be significantly lower than that among the older male subjects [29,30]. Our data revealed that both the prevalence of the equol producers and the daily intake of daidzein calculated based on the BDHQ increase with the age. In the younger Japanese generation, the consumption of soy products has been decreasing with the increase of Western food intake. Decreased intake of daidzein might be associated with lower prevalence of the equol producers among the younger subjects.

Previous studies have suggested that the consumption of isoflavone daidzein does not convert an equol non-producer to an equol producer [13,14]. This concept has been widely accepted, although this has been shown for a relatively shorter duration of daidzein consumption (less than two months) [12,13,14]. In contrast, in an observational study, 18% of the non-producers changed to equol producers after the consumption of isoflavones for years [15]. Another study revealed that 16 weeks exposure to the soy isoflavones changed 8 of 20 (40%) equol non-producers to come equol producers [11]. In addition, a randomized crossover study with a 6-month exposure to soy foods also revealed that 16% of 79 pre-menopausal women acquired the ability to produce equol [31]. Therefore, the duration of the consumption of isoflavone daidzein seems to be important in the acquisition of the ability to produce equol. However, one study on postmenopausal women demonstrated the changes in the fecal bacterial community that contribute to equol production in the non-producers after only one week of isoflavones consumption [32]. The present study showed that some non-producers possess the several bacteria that contribute to equol production. Therefore, among the equol non-producers, the subjects who possessed specific intestinal bacteria in their gut microbiota can transform into equol-producers even after a short period of isoflavone consumption.

Although the bacteria which convert daidzein to equol existed in the most equol non-producer, there were differences in the complexity of the gut microbiota and prevalence of those bacteria between the equol producers and non-producers. Especially, both the relative abundance and the prevalence of *A. celatus* and *S. isoflavoniconvertens* were significantly higher in the equol producers than those in the non-producers. The daily daidzein intake based on the BDHQ analysis was increased with the age. Similarly, in the equol producers, the prevalence of *A. celatus* and *S. isoflavoniconvertens* increased with age. Therefore, daidzein intake would increase these two bacterial species and this increase would play a role in the development of the ability to produce equol.

Several limitations of this study need to be mentioned. Firstly, this study assessed the bacterial species using 16S ribosomal RNA gene sequences, but not by metagenome analysis. Therefore, the strains or functions of the species were not investigated. Second, since the mass-survey was performed in one day, the intake of soy foods for a few days prior to the mass-survey might have influenced the urine equol concentration. However, the BDHQ analysis was performed based on the record of a month prior to the survey, and thus, only 3% of the subjects were recognized as not having consumed daidzein.

In conclusion, our study revealed that not only the equol producers, but also the non-producers possess the bacteria that convert daidzein to equol. Most people in this area located in north Japan possessed the gut microbiota that have been recognized to take part in equol production. However, a higher gut microbiota complexity was observed in the equol producers. Moreover, isoflavone daidzein intake may be associated with the equol production status; particularly, the abundance of *A. celatus* and *S. isoflavoniconvertens* in the gut microbiota may play a significant role in equol production.

## Figures and Tables

**Figure 1 nutrients-11-00433-f001:**
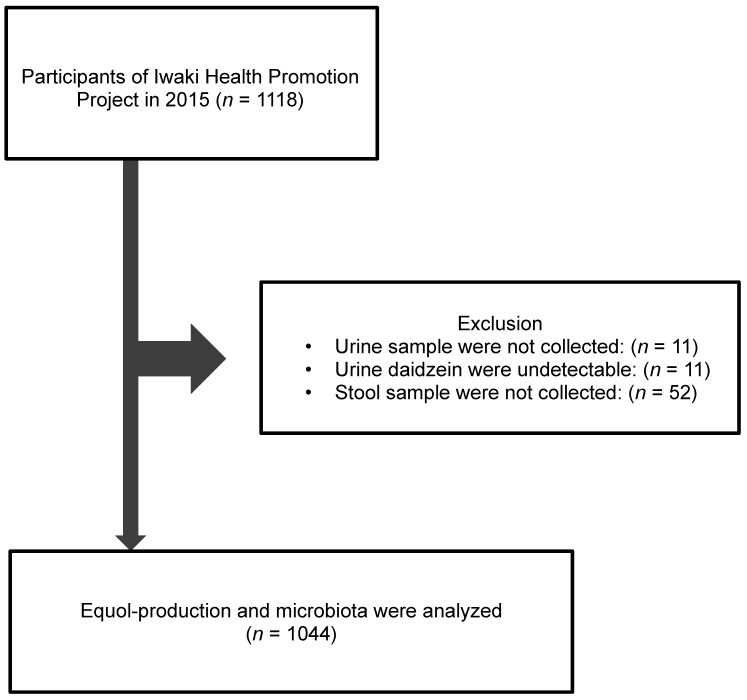
Study flow of subjects. A total 1044 subjects were enrolled from 1118 adults who participated in Iwaki Health Promotion Projects in 2015.

**Figure 2 nutrients-11-00433-f002:**
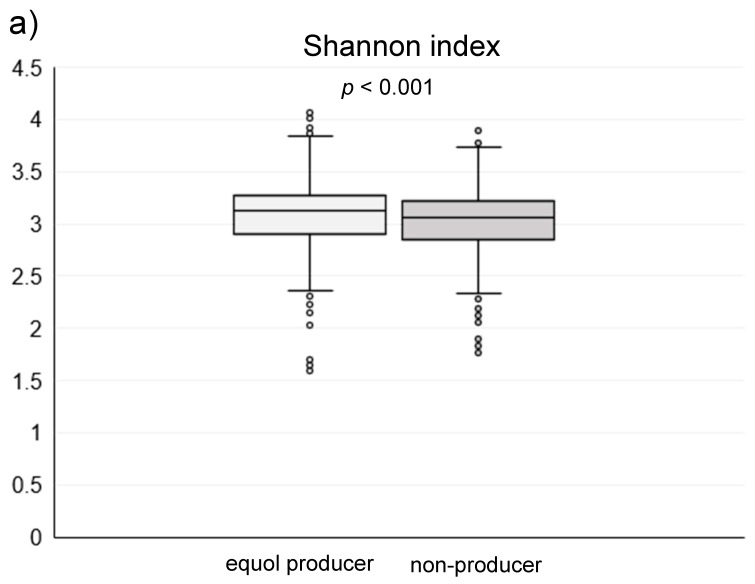
(**a**) The Shannon index and (**b**) Chao1 index of the gut microbiota in the equol producers and non-producers.

**Figure 3 nutrients-11-00433-f003:**
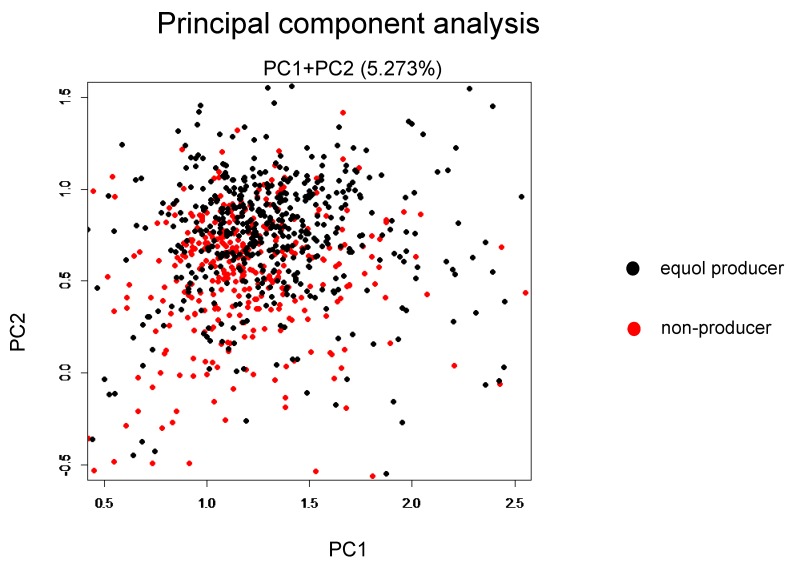
Principal component analysis between the equol producers and non-producers.

**Figure 4 nutrients-11-00433-f004:**
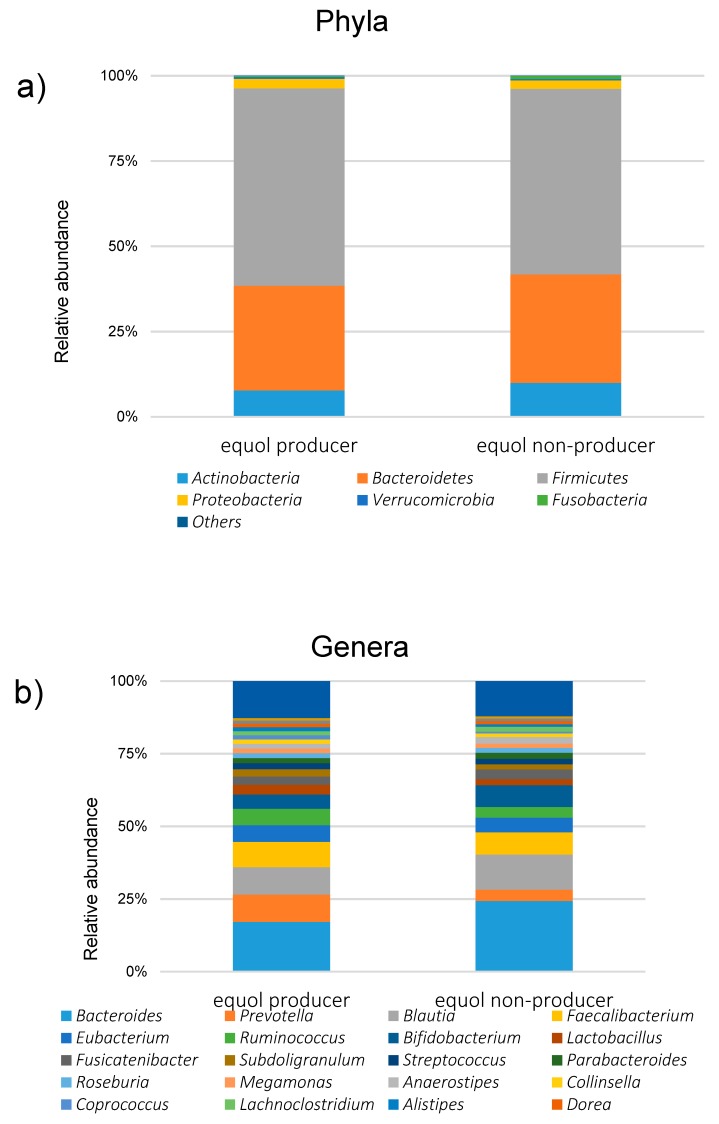
The relative abundances of different gut microbiota between the equol producers and the non-producers (**a**) at the phyla level and (**b**) genera level.

**Figure 5 nutrients-11-00433-f005:**
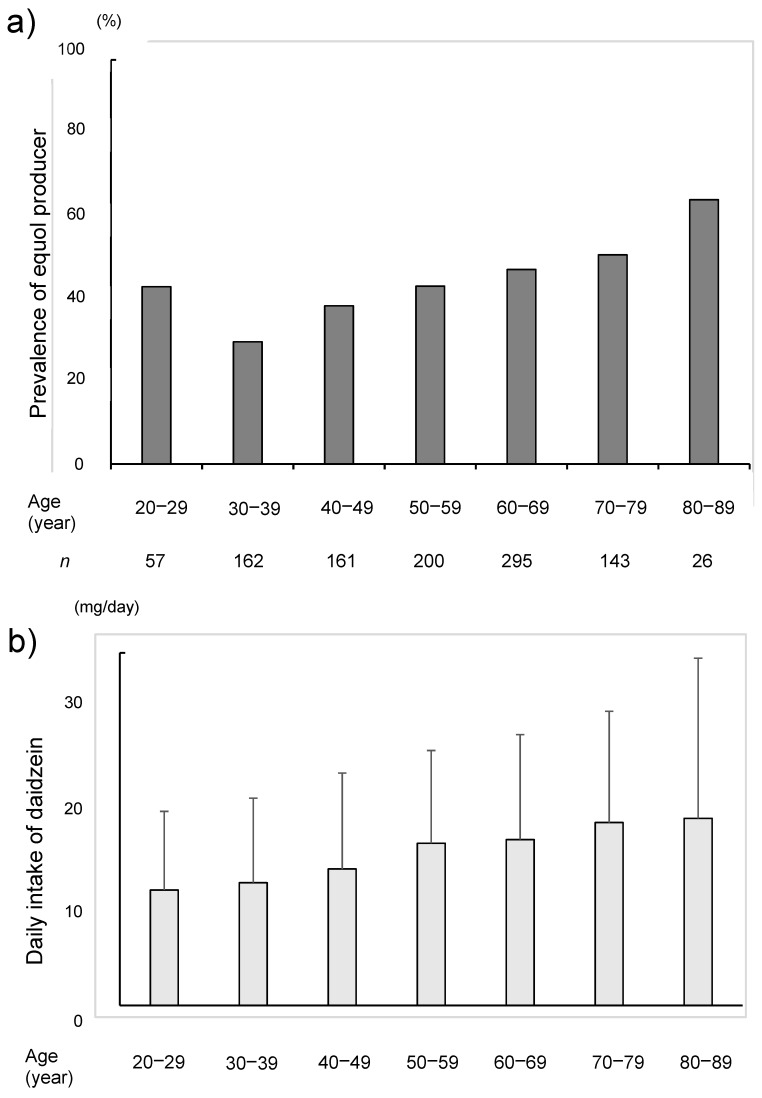
The rate of equol producer (**a**) and the daily intake of daidzein (**b**) among the seven age groups and by decades.

**Figure 6 nutrients-11-00433-f006:**
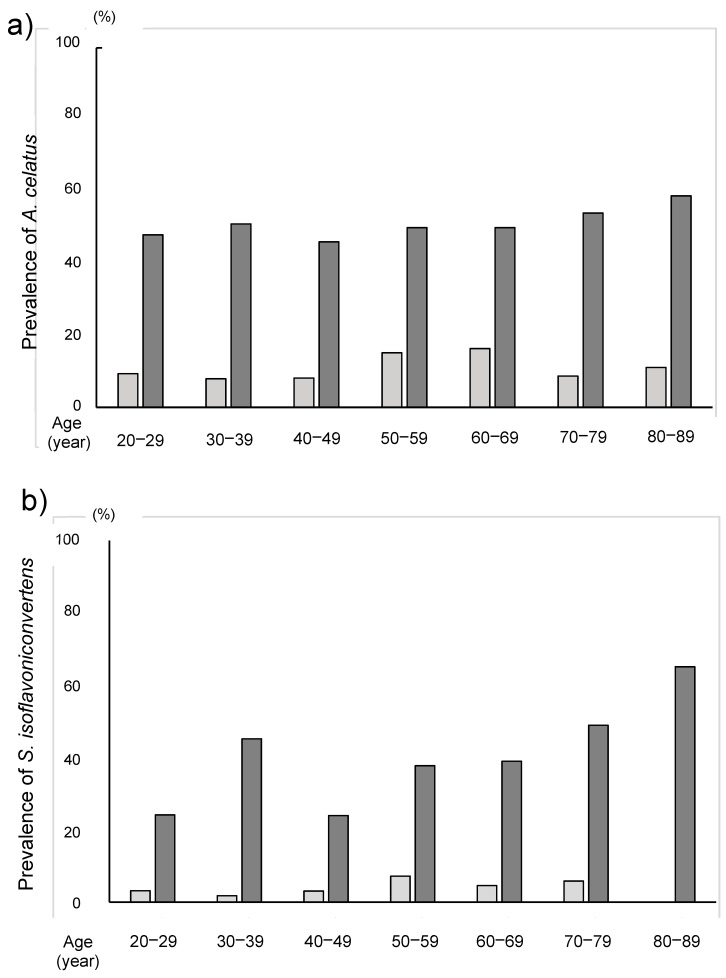
The prevalence of (**a**) *Asaccharobacter celatus* and (**b**) *Slackia isoflavoniconvertens* among the seven age groups by decades.

**Table 1 nutrients-11-00433-t001:** Characteristics of equol producer and non-producer.

	Equol Producer	Non-Producer	*p*-Value
number	458 (43.8%)	586 (56.2%)	
Sex, male	177 (38.6%)	234 (39.9%)	0.673
Age (year)	56.8 ± 14.7	52.9 ± 15.0	<0.001
Body mass index (kg/m^2^)	22.90 ± 3.69	22.56 ± 4.19	0.167
Intake of daidzein (mg/day)	15.0 (13.3)	12.8 (12.3)	0.004
Urine equol concentration (μmol)	6.49 (14.93)	0.04 (0.05)	<0.001
Urine daidzein concentration (μmol)	5.95 (14.68)	12.12 (23.43)	<0.001
log (Urine equol/daidzein)	−0.03 (1.17)	−2.45 (0.69)	<0.001

Data were expressed as mean ± standard deviation or median with interquartile range in parenthesis.

**Table 2 nutrients-11-00433-t002:** Relative abundance of species (percentage of the total bacterial 16sRNA).

	Equol Producer (*n* = 458)	Non-Producer (*n* = 586)	*p* Value	WAD	Rank
*Adlercreutzia equolifaciens*	4.05 ± 32.16	3.78 ± 29.16	0.121	0.000	6
*Asaccharobacter celatus*	1.71 ± 3.52	0.37 ± 14.56	<0.001	0.094	3
*Bacteroides ovatus*	57.34 ± 112.37	117.78 ± 219.26	0.058	0.180	2
*Finegoldia magna*	0.27 ± 1.76	0.30 ± 2.04	0.124	0.001	5
*Lactobacillus mucosae*	3.75 ± 25.66	8.90 ± 83.56	0.481	0.011	4
*Slackia equolifaciens*	0.01 ± 0.06	0	0.808	0	8
*Slackia isoflavoniconvertens*	6.87 ± 15.06	0.72 ± 0.53	<0.001	0.264	1
*Streptococcus intermedius*	0.04 ± 0.17	0.06 ± 0.03	0.860	0.000	7

Data were expressed as (mean ± standard deviation) × 10^4^. WAD—weighted average difference.

**Table 3 nutrients-11-00433-t003:** Prevalence of intestinal bacteria species producing equol.

	Equol Producer		Non-Producer		*p*
*Adlercreutzia equolifaciens*	51	(11.1)	31	(5.3)	<0.001
*Asaccharobacter celatus*	230	(50.2)	69	(11.8)	<0.001
*Bacteroides ovatus*	406	(88.6)	445	(75.9)	<0.001
*Finegoldia magna*	68	(14.8)	120	(20.5)	0.019
*Lactobacillus mucosae*	56	(12.2)	86	(14.7)	0.252
*Slackia equolifaciens*	4	(0.8)	0	(0)	0.023
*Slackia isoflavoniconvertens*	178	(38.9)	25	(4.3)	<0.001
*Streptococcus intermedius*	34	(7.4)	47	(8.0)	0.721

Data were expressed as number with percentage in parenthesis.

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
