# Peer review of "Daidzein Intake Is Associated with Equol Producing Status through an Increase in the Intestinal Bacteria Responsible for Equol Production"

_nutrients, 2019, doi:10.3390/nu11020433_

Reviewer 1 Report

The authors demonstrated the differences of microbiota between the equal producers and the non-producers in the subjects in Hirosaki, Japan. The present study was well organized and will give us a new information especially in the field of the interaction between microbiota and food factors. To improve the quality of this paper, the authors should revise it according to the following suggestions;

1) The authors performed a metagenome analysis by the 16S rRNA V3-V4 squencing in feces samples of 1044 subjects. However, the authors give us the data of 8 representative microbiota. The authors should give us the alpha- and beta-diversity of  between the equal producers and the non-producers, and the abundances pf phylum or genus levels. 

2) The authors obtained differences in 8 representative microbiota between the equal producers and the non-producers, and showed the significant differences in 2 microbiota, and the tendencies of differences in some microbiome. In addition to the statistical significance, the authors should consider the contribution of the abundance of these microbiota. In these case, it would be better to analyze them by the weighted average didderence (WAD) algorithum using R statistical software. 

3) The authors obtained the data from BDHQ. The authors should analyze the correlation between the BDHQ data and the abundance of 8 representative microbiota, and should show them by the heat-map data. 

Author Response

1) The authors performed a metagenome analysis by the 16S rRNA V3-V4 sequencing in feces samples of 1044 subjects. However, the authors give us the data of 8 representative microbiota. The authors should give us the alpha- and beta-diversity of between the equal producers and the non-producers, and the abundances pf phylum or genus levels.

Reply: Thank you for the comments. According to the reviewer’s comments, we performed the alpha- and beta-diversity analysis of the gut microbiota of the equol producers and non-producers. There were significant differences and we showed the results in Figure 2 and 3, respectively. Moreover, we also assessed the relative abundance of the gut microbiota at phylum and genus levels and the results are shown in Figure 4

2) The authors obtained differences in 8 representative microbiota between the equal producers and the non-producers, and showed the significant differences in 2 microbiota, and the tendencies of differences in some microbiome. In addition to the statistical significance, the authors should consider the contribution of the abundance of these microbiota. In these case, it would be better to analyze them by the weighted average didderence (WAD) algorithum using R statistical software.

Reply: As reviewer’s comment, we analyzed the contribution of the abundance of 8 representative microbiota by the weighted average difference and added the results to Table 2.

3) The authors obtained the data from BDHQ. The authors should analyze the correlation between the BDHQ data and the abundance of 8 representative microbiota, and should show them by the heat-map data.

Reply: We analyzed the correlation between the BDHQ data the abundance of 8 representative microbiota. However, there were no significant correlation between the amount of soy foods and the abundance of 8 representative microbiota in both groups. Therefore, we could not make significant heat-map.

Reviewer 2 Report

This manuscript is an excellent paper written from an interesting aspect of equol producing. I have suggestions for the authors to consider. Please answer if possible.

Although there is a significant difference in daidzein intake between equol producer and non-producer, it is not a decisive difference. Is there daidzein intake necessary to change from equol non-producer to producer?

Whether features are recognized in the type of soybean food ingested between equol producer and non-producer?

Author Response

Although there is a significant difference in daidzein intake between equol producer and non-producer, it is not a decisive difference. Is there daidzein intake necessary to change from equol non-producer to producer?

Reply: Thank you very much for the comments. We believe that daidzein intake is necessary to change equol non-producer to producer. However, this study is retrospective and thus we cannot answer the question. Prospective study is necessary for this question.

Whether features are recognized in the type of soybean food ingested between equol producer and non-producer?

Reply: In BDHQ, intake of three types of soybean foods (natto, tofu, and fried tofu) was recorded. We described about this in the method section. As described in the previous manuscript, daily intake of daidzein was higher in equol-producer. However, daily intake of individual soy food was not significantly different between the equol producer and non-producer.

Round  2

Reviewer 1 Report

The authors revised it according to the suggestions. We have no claim in the revised paper.